# Genetic Algebras Associated with *ξ*^(*a*)^-Quadratic Stochastic Operators

**DOI:** 10.3390/e25060934

**Published:** 2023-06-13

**Authors:** Farrukh Mukhamedov, Izzat Qaralleh, Taimun Qaisar, Mahmoud Alhaj Hasan

**Affiliations:** 1Department of Mathematical Sciences, College of Science, United Arab Emirates University, Al Ain P.O. Box 15551, United Arab Emirates or taimunqaisar@gmail.com (T.Q.); 202050061@uaeu.ac.ae (M.A.H.); 2Department of Mathematics, Tafila Technical University, Tafila P.O. Box 66110, Jordan; izzat_math@ttu.edu.jo

**Keywords:** quadratic stochastic operator, associativity, dynamics

## Abstract

The present paper deals with a class of $\xi^{\left(a\right)}$-quadratic stochastic operators, referred to as QSOs, on a two-dimensional simplex. It investigates the algebraic properties of the genetic algebras associated with $\xi^{\left(a\right)}$-QSOs. Namely, the associativity, characters and derivations of genetic algebras are studied. Moreover, the dynamics of these operators are also explored. Specifically, we focus on a particular partition that results in nine classes, which are further reduced to three nonconjugate classes. Each class gives rise to a genetic algebra denoted as $A_{i}$, and it is shown that these algebras are isomorphic. The investigation then delves into analyzing various algebraic properties within these genetic algebras, such as associativity, characters, and derivations. The conditions for associativity and character behavior are provided. Furthermore, a comprehensive analysis of the dynamic behavior of these operators is conducted.

## 1. Introduction

Mathematical population genetics investigates the dynamics of frequency distributions of genetic types (alleles, genotypes, gene collections, etc.) in successive generations under the action of evolutionary forces. To explore the behavior of the population, the discrete dynamical system associated with an evolution operator is the main object of the theory. In ref. [1], a short history of applications of mathematics to solving various problems in population dynamics is given.

On the other hand, there is another theoretical framework to investigate essential properties of population genetics, which is based on an algebraic approach [2,3]. In this scheme, most of the algebras are nonassociative. In the literature (see, for example, [4,5]), plenty of nonassociative algebras (baric, evolution, Bernstein, train, stochastic, etc.) have appeared to model inheritance in genetic systems. Such algebras are referred to as “genetic algebras”. In general, problems of population genetics were started in [6] by employing quadratic stochastic operators (see also [2]). It is worth recalling those operators which present the time evolution of species in biology [7,8]. Namely, let us look at a population consisting of *m* species (or traits) which are denoted by I={1,2,⋯,m}. Assume that $x^{\left(0\right)}=\left(x_{1}^{\left(0\right)},\cdots ,x_{m}^{\left(0\right)}\right)$ is a probability distribution of species at an initial state, and $p_{ij,k}$ is a probability that individuals in the *i*th and *j*th species interbreed to produce an individual from a *k*th species. Then, a probability distribution $x^{\left(1\right)}=\left(x_{1}^{\left(1\right)},\cdots ,x_{m}^{\left(1\right)}\right)$ of the species in the first generation can be found as a total probability, i.e.,
$$x_{k}^{\left(1\right)}=\sum_{i,j=1}^{m}p_{ij,k}x_{i}^{\left(0\right)}x_{j}^{\left(0\right)},\;k\in \left\{1,\ldots ,m\right\}.$$ The correspondence $x^{\left(0\right)}\to x^{\left(1\right)}$ is called *the evolution operator* or *quadratic stochastic operator* (QSO). In other words, such an operator describes a distribution of the next generation if the distribution of the current generation was given. Applications of QSOs to population genetics were given in [2,9,10,11]. The reader is referred to [12] for a self-contained exposition of the recent achievements and open problems in the theory of QSOs.

QSOs find applications as discrete-time dynamical systems in various fields, including economics, epidemiology, and social sciences [13,14,15,16,17]. They are valuable tools for analyzing and predicting the behavior of complex systems that undergo discrete changes at fixed time intervals. For instance, in economics, QSOs can assist in modeling and understanding market dynamics, optimizing resource allocation, and predicting economic trends. In epidemiology, QSOs can be employed to simulate disease spread, evaluate intervention strategies, and forecast the progression of infectious diseases. Moreover, in social sciences, QSOs can aid in studying social dynamics, opinion formation, and decision-making processes within populations. By employing QSOs as discrete-time dynamical systems, researchers can gain insights into the intricate dynamics of these complex systems, enabling better understanding, planning, and decision-making.

Each QSO defines an algebraic structure on the vector space $R^{m}$ containing the simplex (see next section for definitions). The associated algebra is called *genetic algebra*. A more modern use of the genetic algebra theory for self-fertilization can be found in [18,19]. Therefore, it is the interplay between the purely mathematical structure and the corresponding genetic properties that makes this subject so fascinating. We refer to [2,3,20] for comprehensive references.

In [21,22], new classes of QSO were introduced, which are called $\xi^{\left(as\right)}$-QSOs. We notice that such classes of operators depend on the partition of the coupled index set (the coupled trait set) $P_{m}=\left\{\left(i,j\right):i<j\right\}\subset I\times I$. Furthermore, certain subclasses of these operators have been intensively explored in [23,24,25]. However, in those investigations, the algebraic structures of genetic algebras associated with $\xi^{\left(a\right)}$-QSO are not considered. Therefore, to fill that gap, in the present paper, we are aiming to study certain algebraic properties of genetic algebras corresponding to $\xi^{\left(a\right)}$-QSO. We stress that the considered $\xi^{\left(a\right)}$-QSOs are different from Lotka–Volterra QSOs, which also have important applications in several branches of sciences [26,27,28,29]. The genetic algebras associated with Lotka–Volterra operators have been intensively explored in [30,31,32,33]. There appeared several works on the derivations of genetic algebras [34,35,36,37]. Interpretations of the derivations have been discussed in [18]. Recently, in [38,39,40,41], derivations of Lotka–Volterra algebras have been described. Furthermore, other types of genetic algebras have been investigated in [42,43,44,45,46,47].

The paper is organized as follows. In Section 2, we collect necessary definitions from the theory of genetic algebras. Section 3 is devoted to the construction of a class of $\xi^{\left(a\right)}$-QSO on two-dimensional simplex. Furthermore, in Section 4, we study the associativity of these operators along with their dynamics. The characters of these algebras are described in Section 5. In Section 6, the derivations of genetic algebras associated with $\xi^{\left(a\right)}$ are described. Moreover, in Section 7, the dynamics of these operators are discussed.

## 2. Preliminaries

Recall that a quadratic stochastic operator (QSO) is a mapping of the simplex
(1)$$\begin{array}{c} S^{m-1}=\left\{x=\left(x_{1},\cdots ,x_{m}\right)\in R^{m}:\;\;\sum_{i=1}^{m}x_{i}=1,\;\;x_{i}\geq 0,\;\;i=\bar{1,m}\right\} \end{array}$$
into itself, of the form
(2)$$\begin{array}{c} x_{k}^{\prime }=\sum_{i,j=1}^{m}P_{ij,k}x_{i}x_{j},\;\;\;k=\bar{1,m}, \end{array}$$
where $V\left(x\right)=x^{\prime }=\left(x_{1}^{\prime },\cdots ,x_{m}^{\prime }\right)$, and $P_{ij,k}$ is a coefficient of heredity, which satisfies the following conditions
(3)$$\begin{array}{c} P_{ij,k}\geq 0,\;P_{ij,k}=P_{ji,k},\;\sum_{k=1}^{m}P_{ij,k}=1. \end{array}$$ Thus, each quadratic stochastic operator $V:S^{m-1}\to S^{m-1}$ can be uniquely defined by a cubic matrix $P={\left(P_{ijk}\right)}_{i,j,k=1}^{m}$ with conditions (Equation 3).

A point $x\in S^{m-1}$ is called a *k*-periodic point of *V*, if $V^{j}\left(x\right)\neq x$, 0≤j<k, $V^{k}\left(x\right)=x$. If k=1, then such a point is called a fixed point of *V*. The set of fixed points and k− periodic points of *V* are denoted by Fix(V) and $Per_{k}\left(V\right)$, respectively. For a given point $x^{\left(0\right)}\in S^{m-1}$, a trajectory ${\left\{x^{\left(n\right)}\right\}}_{n=0}^{\infty }$ of *V* starting from $x^{\left(0\right)}$ is defined by $x^{\left(n+1\right)}=V\left(x^{\left(n\right)}\right)$. By $\omega_{V}\left(x^{\left(0\right)}\right)$, we denote a set of omega limiting points of the trajectory ${\left\{x^{\left(n\right)}\right\}}_{n=0}^{\infty }$.

**Definition 1.** 
*A quadratic stochastic operator V is called regular if for any initial point $x\in S^{m-1}$ the limit $\lim_{n\to \infty }V^{n}\left(x\right)$ exists.*


Note that each element $x\in S^{m-1}$ is a probability distribution of the set I={1,…,m}. Let $x=(x_{1},\cdots ,x_{m})$ and $y=(y_{1},\cdots ,y_{m})$ be vectors taken from $S^{m-1}$. We say that *x is equivalent to y* if $x_{k}=0$⇔$y_{k}=0$; this relation is denoted by x∼y.

Let $supp\left(x\right)=\left\{i:x_{i}\neq 0\right\}$ be a support of $x\in S^{m-1}$. We say that *x is singular to y* and denote by x⊥y, if supp(x)∩supp(y)=∅.

We denote the sets of coupled indexes by
$$P_{m}=\left\{\left(i,j\right):\;i<j\right\}\subset I\times I,\;\Delta_{m}=\left\{\left(i,i\right):i\in I\right\}\subset I\times I.$$ For a given pair $\left(i,j\right)\in P_{m}\cup \Delta_{m}$, we set a vector $P_{ij}=\left(P_{ij,1},\cdots ,P_{ij,m}\right)$. It is clear due to condition (Equation 3) that $P_{ij}\in S^{m-1}$.

Let $\xi ={\left\{A_{i}\right\}}_{i=1}^{N}$ and $\eta ={\left\{B_{i}\right\}}_{i=1}^{M}$ be some fixed partitions of $P_{m}$ and $\Delta_{m}$, respectively, i.e., $A_{i}⋂A_{j}=\emptyset$, $B_{i}⋂B_{j}=\emptyset$, and $\overset{N}{\underset{i=1}{⋃}}A_{i}=P_{m}$, $\overset{M}{\underset{i=1}{⋃}}B_{i}=\Delta_{m}$, where N,M≤m.

**Definition 2.** 
*A quadratic stochastic operator $V:S^{m-1}\to S^{m-1}$ given by (Equation 2) and (Equation 3), is called a $\xi^{\left(as\right)}$-QSO with regard to the partitions ξ,η (where the letters “as” stand for absolutely continuous-singular) if the following conditions are satisfied:*
*(i)* 
*for each k∈{1,…,N} and any (i,j), $\left(u,v\right)\in A_{k}$, one has that $P_{ij}∼P_{uv}$;*
*(ii)* 
*for any k≠ℓ, k,ℓ∈{1,…,N} and any $\left(i,j\right)\in A_{k}$ and $\left(u,v\right)\in A_{\ell }$ one has that $P_{ij}⊥P_{uv}$;*
*(iii)* 
*for each d∈{1,…,M} and any (i,i), $\left(j,j\right)\in B_{d}$, one has that $P_{ii}∼P_{jj}$;*
*(iv)* 
*for any s≠h, s,h∈{1,…,M} and any $\left(u,u\right)\in B_{s}$ and $\left(v,v\right)\in B_{h}$ one has that $P_{uu}⊥P_{vv}$.*



**Remark 1.** 
*If η is the point partition, i.e. $\xi_{2}=\left\{\left\{\left(1,1\right)\right\},\ldots \left\{\left(m,m\right)\right\}\right\}$, then we call the corresponding QSO by $\xi^{\left(s\right)}$-QSO (where the letter “s” stands for singularity), because in this case, every two different vectors $P_{ii}$ and $P_{jj}$ are singular. If η is the trivial, i.e., $\xi_{2}=\left\{\Delta_{m}\right\}$, then we call the corresponding QSO by $\xi^{\left(a\right)}$-QSO (where the letter “a” stands for absolute continuous), because in this case, every two vectors $P_{ii}$ and $P_{jj}$ are equivalent. We note that some classes of $\xi^{\left(a\right)}$-QSO have been studied in [21].*


A BIOLOGICAL INTERPRETATION OF A $\xi^{\left(a\right)}-$ QSO: We treat I={1,⋯,m} as a set of all possible traits of the population system. A coefficient $P_{ij,k}$ is a probability that parents in the $i^{th}$ and $j^{th}$ traits interbreed to produce a child from the $k^{th}$ trait. The condition $P_{ij,k}=P_{ji,k}$ means that the gender of parents does not influence the probability of having a child from the $k^{th}$ trait. In this sense, $P_{m}\cup \Delta_{m}$ is a set of all the possible coupled traits of the parents. A vector $P_{ij}=\left(P_{ij,1},\cdots ,P_{ij,m}\right)$ is a possible distribution of children in a family where the parents are carrying traits from the $i^{th}$ and $j^{th}$ types.

## 3. A Class of $\xi^{\left(a\right)}$-QSO on 2D Simplex

In this section, we are going to define $\xi^{\left(a\right)}$-QSO in two-dimensional simplex, i.e., m=3. The set $P_{3}$ has the following possible partitions:$$\begin{array}{ccc} & & \xi_{1}=\left\{\left\{\left(1,2\right)\right\},\left\{\left(1,3\right)\right\},\left\{\left(2,3\right)\right\}\right\},\;\left|\xi_{1}\right|=3,\\ & & \xi_{2}=\left\{\left\{\left(2,3\right)\right\},\left\{\left(1,2\right),\left(1,3\right)\right\}\right\},\;\left|\xi_{2}\right|=2,\\ & & \xi_{3}=\left\{\left\{\left(1,3\right)\right\},\left\{\left(1,2\right),\left(2,3\right)\right\}\right\},\;\left|\xi_{3}\right|=2,\\ & & \xi_{4}=\left\{\left\{\left(1,2\right)\right\},\left\{\left(1,3\right),\left(2,3\right)\right\}\right\},\;\left|\xi_{4}\right|=2,\\ & & \xi_{5}=\left\{\left(1,2\right),\left(1,3\right),\left(2,3\right)\right\},\;\left|\xi_{5}\right|=1. \end{array}$$

We notice that $\xi^{\left(as\right)}$-QSOs corresponding to the partitions $\xi_{1}-\xi_{4}$ have been studied in [22,23,24,25]. Therefore, in the present paper, we concentrate on the partition $\xi_{5}$ and η={(1,1)(2,2)(3,3)}, which defines a class of $\xi^{\left(a\right)}$-QSO. In the sequel, for the sake of simplicity, we are going to consider the following coefficients ${\left(P_{ij,k}\right)}_{i,j,k=1}^{m}$ given by the table:
$P_{11}$$P_{22}$$P_{33}$$P_{12}$$P_{13}$$P_{23}$(α,1−α,0)(1−α,α,0)(1−α,α,0)(1,0,0)(1,0,0)(1,0,0)(α,0,1−α)(1−α,0,α)(1−α,0,α)(0,1,0)(0,1,0)(0,1,0)(0,α,1−α)(0,1−α,α)(0,1−α,α)(0,0,1)(0,0,1)(0,0,1) where α∈[0,1].

The corresponding QSOs are listed as follows: (4)$$V_{1}:=\left\{\begin{array}{c} x_{1}^{{}^{\prime }}=\alpha x_{1}^{2}+\left(1-\alpha \right)x_{2}^{2}+\left(1-\alpha \right)x_{3}^{2}+2x_{1}x_{2}+2x_{2}x_{3}+2x_{1}x_{3}\\ x_{2}^{{}^{\prime }}=\left(1-\alpha \right)x_{1}^{2}+\alpha x_{2}^{2}+\alpha x_{3}^{2}\\ x_{3}^{{}^{\prime }}=0 \end{array}\right.$$



(5)
$$V_{2}:=\left\{\begin{array}{c} x_{1}^{{}^{\prime }}=\alpha x_{1}^{2}+\left(1-\alpha \right)x_{2}^{2}+\left(1-\alpha \right)x_{3}^{2}+2x_{1}x_{2}+2x_{2}x_{3}+2x_{1}x_{3}\\ x_{2}^{{}^{\prime }}=0\\ x_{3}^{{}^{\prime }}=\left(1-\alpha \right)x_{1}^{2}+\alpha x_{2}^{2}+\alpha x_{3}^{2} \end{array}\right.$$


(6)
$$V_{3}:=\left\{\begin{array}{c} x_{1}^{{}^{\prime }}=2x_{1}x_{2}+2x_{2}x_{3}+2x_{1}x_{3}\\ x_{2}^{{}^{\prime }}=\alpha x_{1}^{2}+\left(1-\alpha \right)x_{2}^{2}+\left(1-\alpha \right)x_{3}^{2}\\ x_{3}^{{}^{\prime }}=\left(1-\alpha \right)x_{1}^{2}+\alpha x_{2}^{2}+\alpha x_{3}^{2} \end{array}\right.$$


(7)
$$V_{4}:=\left\{\begin{array}{c} x_{1}^{{}^{\prime }}=\alpha x_{1}^{2}+\left(1-\alpha \right)x_{2}^{2}+\left(1-\alpha \right)x_{3}^{2}\\ x_{2}^{{}^{\prime }}=\left(1-\alpha \right)x_{1}^{2}+\alpha x_{2}^{2}+\alpha x_{3}^{2}+2x_{1}x_{2}+2x_{2}x_{3}+2x_{1}x_{3}\\ x_{3}^{{}^{\prime }}=0 \end{array}\right.$$


(8)
$$V_{5}:=\left\{\begin{array}{c} x_{1}^{{}^{\prime }}=\alpha x_{1}^{2}+\left(1-\alpha \right)x_{2}^{2}+\left(1-\alpha \right)x_{3}^{2}\\ x_{2}^{{}^{\prime }}=2x_{1}x_{2}+2x_{2}x_{3}+2x_{1}x_{3}\\ x_{3}^{{}^{\prime }}=\left(1-\alpha \right)x_{1}^{2}+\alpha x_{2}^{2}+\alpha x_{3}^{2} \end{array}\right.$$


(9)
$$V_{6}:=\left\{\begin{array}{c} x_{1}^{{}^{\prime }}=0\\ x_{2}^{{}^{\prime }}=\alpha x_{1}^{2}+\left(1-\alpha \right)x_{2}^{2}+\left(1-\alpha \right)x_{3}^{2}+2x_{1}x_{2}+2x_{2}x_{3}+2x_{1}x_{3}\\ x_{3}^{{}^{\prime }}=\left(1-\alpha \right)x_{1}^{2}+\alpha x_{2}^{2}+\alpha x_{3}^{2} \end{array}\right.$$


(10)
$$V_{7}:=\left\{\begin{array}{c} x_{1}^{{}^{\prime }}=\alpha x_{1}^{2}+\left(1-\alpha \right)x_{2}^{2}+\left(1-\alpha \right)x_{3}^{2}\\ x_{2}^{{}^{\prime }}=\left(1-\alpha \right)x_{1}^{2}+\alpha x_{2}^{2}+\alpha x_{3}^{2}\\ x_{3}^{{}^{\prime }}=2x_{1}x_{2}+2x_{2}x_{3}+2x_{1}x_{3} \end{array}\right.$$


(11)
$$V_{8}:=\left\{\begin{array}{c} x_{1}^{{}^{\prime }}=\alpha x_{1}^{2}+\left(1-\alpha \right)x_{2}^{2}+\left(1-\alpha \right)x_{3}^{2}+\\ x_{2}^{{}^{\prime }}=0\\ x_{3}^{{}^{\prime }}=\left(1-\alpha \right)x_{1}^{2}+\alpha x_{2}^{2}+\alpha x_{3}^{2}+2x_{1}x_{2}+2x_{2}x_{3}+2x_{1}x_{3} \end{array}\right.$$


(12)
$$V_{9}:=\left\{\begin{array}{c} x_{1}^{{}^{\prime }}=0\\ x_{2}^{{}^{\prime }}=\alpha x_{1}^{2}+\left(1-\alpha \right)x_{2}^{2}+\left(1-\alpha \right)x_{3}^{2}\\ x_{3}^{{}^{\prime }}=\left(1-\alpha \right)x_{1}^{2}+\alpha x_{2}^{2}+\alpha x_{3}^{2}+2x_{1}x_{2}+2x_{2}x_{3}+2x_{1}x_{3} \end{array}\right.$$



## 4. Associativity

Let *V* be a QSO, and suppose that $x,y\in R^{m}$ are arbitrary vectors. Then, one can define a binary rule [9] on $R^{m}$ by
(13)$${\left(x\circ_{V}y\right)}_{k}=\sum_{i,j=1}^{m}P_{ij,k}x_{i}x_{j}.$$ Using (Equation 3), one can see that $x\circ_{V}y=y\circ_{V}x$, i.e., the multiplication is commutative. Certain algebraic properties of such kinds of algebras were investigated in [2,3,20]. In general, genetic algebra is not necessarily associative.

The multiplication (Equation 13) in the canonical basis can be represented as follows:(14)$$e_{i}\circ_{V}e_{j}=\sum_{i,j=1}^{m}P_{ij,k}e_{k}.$$ It turns out that the multiplication can be given terms of QSO
$$x\circ_{V}y=\frac{1}{4}\left(V\left(x+y\right)-V\left(x-y\right)\right).$$ One can check that
$$x\circ_{V}x=x^{2}=V\left(x\right)\;\;\mathrm{for}\;\mathrm{any}\;x\in S^{m-1}.$$ This algebraic interpretation is useful, e.g., a state x is an equilibrium precisely when x is an idempotent element of the algebra.

The algebra *A* is called *associative* if
(x∘y)∘z=x∘(y∘z)for allx,y,z,∈A.

In this section, we are going to investigate the associativity of genetic algebras generated by $\xi^{\left(a\right)}$-QSO described in the previous section. To describe such algebras, we are going to consider more general operators which cover all listed ones. For this reason, we are going to evaluate the following table:
$P_{11,1}=a_{1}\;$$P_{11,2}=b_{1}\;$$P_{11,3}=c_{1}\;$$P_{22,1}=a_{2}\;$$P_{22,2}=b_{2}\;$$P_{22,3}=c_{2}\;$$P_{33,1}=a_{3}\;$$P_{33,2}=b_{3}\;$$P_{33,3}=c_{3}\;$ where $a_{i},b_{i},c_{i}\geq 0$
$$a_{1}+b_{1}+c_{1}=1,\;\;a_{2}+b_{2}+c_{2}=1,\;\;a_{3}+b_{3}+c_{3}=1.\;$$ Furthermore, we assume that the coefficients ${\left(P_{ij,k}\right)}_{ij,k=1}^{m}$ are given by
Cases$P_{12}$$P_{13}$$P_{23}$1(1,0,0)(1,0,0)(1,0,0)2(0,1,0)(0,1,0)(0,1,0)3(0,0,1)(0,0,1)(0,0,1)
Then, the corresponding QSOs are described as follows: (15)$$W_{1}:=\left\{\begin{array}{c} x_{1}^{{}^{\prime }}=a_{1}x_{1}^{2}+a_{2}x_{2}^{2}+a_{3}x_{3}^{2}+2x_{1}x_{2}+2x_{2}x_{3}+2x_{1}x_{3}\\ x_{2}^{{}^{\prime }}=b_{1}x_{1}^{2}+b_{2}x_{2}^{2}+b_{3}x_{3}^{2}\\ x_{3}^{{}^{\prime }}=c_{1}x_{1}^{2}+c_{2}x_{2}^{2}+c_{3}x_{3}^{2} \end{array}\right.$$
(16)$$W_{2}:=\left\{\begin{array}{c} x_{1}^{{}^{\prime }}=a_{1}x_{1}^{2}+a_{2}x_{2}^{2}+a_{3}x_{3}^{2}\\ x_{2}^{{}^{\prime }}=b_{1}x_{1}^{2}+b_{2}x_{2}^{2}+b_{3}x_{3}^{2}+2x_{1}x_{2}+2x_{2}x_{3}+2x_{1}x_{3}\\ x_{3}^{{}^{\prime }}=c_{1}x_{1}^{2}+c_{2}x_{2}^{2}+c_{3}x_{3}^{2} \end{array}\right.$$
(17)$$W_{3}:=\left\{\begin{array}{c} x_{1}^{{}^{\prime }}=a_{1}x_{1}^{2}+a_{2}x_{2}^{2}+a_{3}x_{3}^{2}\\ x_{2}^{{}^{\prime }}=b_{1}x_{1}^{2}+b_{2}x_{2}^{2}+b_{3}x_{3}^{2}\\ x_{3}^{{}^{\prime }}=c_{1}x_{1}^{2}+c_{2}x_{2}^{2}+c_{3}x_{3}^{2}+2x_{1}x_{2}+2x_{2}x_{3}+2x_{1}x_{3} \end{array}\right.$$

The obtained operators $W_{1},W_{2}$ and $W_{3}$, according to (Equation 13), generate corresponding genetic algebras which are denoted by $A_{1},A_{2}$ and $A_{3}$. Therefore, we are going to investigate the associativity of these algebras. Let us list their table of multiplication.

**Case I:** In this case, we consider the QSO $W_{1}$; then, for the corresponding genetic algebra $A_{1}$, the table of multiplication is given by

$e_{1}$$e_{2}$$e_{3}$$e_{1}$$a_{1}e_{1}+b_{1}e_{2}+c_{1}e_{3}$$e_{1}$$e_{1}$$e_{2}$$e_{1}$$a_{2}e_{1}+b_{2}e_{2}+c_{2}e_{3}$$e_{1}$$e_{3}$$e_{1}$$e_{1}$$a_{3}e_{1}+b_{3}e_{2}+c_{3}e_{3}$

**Case II:** Now, let us consider $W_{2}$, then the algebra $A_{2}$ has the following table of multiplication:

$e_{1}$$e_{2}$$e_{3}$$e_{1}$$a_{1}e_{1}+b_{1}e_{2}+c_{1}e_{3}$$e_{2}$$e_{2}$$e_{2}$$e_{2}$$a_{2}e_{1}+b_{2}e_{2}+c_{2}e_{3}$$e_{2}$$e_{3}$$e_{2}$$e_{2}$$a_{3}e_{1}+b_{3}e_{2}+c_{3}e_{3}$

**Case III:** Using the same argument, the algebra $A_{3}$ is defined by $W_{3}$, and its table of multiplication is given by

$e_{1}$$e_{2}$$e_{3}$$e_{1}$$a_{1}e_{1}+b_{1}e_{2}+c_{1}e_{3}$$e_{3}$$e_{3}$$e_{2}$$e_{3}$$a_{2}e_{1}+b_{2}e_{2}+c_{2}e_{3}$$e_{3}$$e_{3}$$e_{3}$$e_{3}$$a_{3}e_{1}+b_{3}e_{2}+c_{3}e_{3}$

**Theorem 1.** 
*The algebras $A_{1},A_{2}$ and $A_{3}$ are isomorphic.*


**Proof.** Let $W_{1}$ be given by (Equation 15) with the parameters $a_{i},b_{i},c_{i}$, and $W_{2}$ be given by (Equation 16) with the following parameters ${\bar{a}}_{i},{\bar{b}}_{i},{\bar{c}}_{i}$, such that
$$\begin{array}{c} \bar{a_{2}}=b_{1},\bar{b_{2}}=a_{1},\bar{c_{2}}=c_{1}\\ \bar{a_{1}}=b_{2},\bar{b_{1}}=a_{2},\bar{c_{1}}=c_{2}\\ \bar{a_{3}}=b_{3},\bar{b_{3}}=a_{3},\bar{c_{3}}=c_{3} \end{array}$$ For the sake of simplicity, we prove that $A_{1}$ is isomorphic to $A_{2}$. To do so, let us define a mapping
$$\alpha \left(x_{1},x_{2},x_{3}\right)=\left(x_{2},x_{1},x_{3}\right).$$ It is enough to check
$$\alpha \left(e_{i}\circ_{W_{1}}e_{j}\right)=\alpha \left(e_{i}\right)\circ_{W_{2}}\alpha \left(e_{j}\right),\;\;\forall i,j\in \left\{1,2,3\right\}.$$ Using Case I and Case II, we find
$$\begin{array}{c} \alpha \left(e_{1}\circ_{W_{1}}e_{1}\right)=\alpha \left(e_{1}\right)\circ_{W_{2}}\alpha \left(e_{1}\right)\Rightarrow \bar{a_{2}}=b_{1},\bar{b_{2}}=a_{1},\bar{c_{2}}=c_{1}\\ \alpha \left(e_{2}\circ_{W_{1}}e_{2}\right)=\alpha \left(e_{2}\right)\circ_{W_{2}}\alpha \left(e_{2}\right)\Rightarrow \bar{a_{1}}=b_{2},\bar{b_{1}}=a_{2},\bar{c_{1}}=c_{2}\\ \alpha \left(e_{3}\circ_{W_{1}}e_{3}\right)=\alpha \left(e_{3}\right)\circ_{W_{2}}\alpha \left(e_{3}\right)\Rightarrow \bar{a_{3}}=b_{3},\bar{b_{3}}=a_{3},\bar{c_{3}}=c_{3}\\ \alpha \left(e_{1}\circ_{W_{1}}e_{2}\right)=\alpha \left(e_{1}\right)\circ_{W_{2}}\alpha \left(e_{2}\right)=e_{2}\\ \alpha \left(e_{1}\circ_{W_{1}}e_{3}\right)=\alpha \left(e_{1}\right)\circ_{W_{2}}\alpha \left(e_{3}\right)=e_{2}\\ \alpha \left(e_{2}\circ_{W_{1}}e_{3}\right)=\alpha \left(e_{2}\right)\circ_{W_{2}}\alpha \left(e_{3}\right)=e_{2} \end{array}$$
which completes the proof. □

Furthermore, due to the proved theorem, we always consider the genetic algebra $A_{1}$.

**Theorem 2.** 
*The genetic algebra $A_{1}$ is associative if and only if one of the following conditions is satisfied.*
*(i)* 
*

$a_{1}=1,b_{1}=0,c_{1}=0\;;a_{2}=1,b_{2}=0,c_{2}=0\;;a_{3},b_{3},c_{3}-arbitary,b_{3}\neq 0$

*
*(ii)* 
*

$a_{1}=1,b_{1}=0,c_{1}=0\;;a_{2},b_{2},c_{2}-arbitary,c_{2}\neq 0\;;a_{3}=1,b_{3}=0,c_{3}=0$

*
*(iii)* 
*

$a_{1}=1,b_{1}=0,c_{1}=0\;;a_{2}-arbitary,b_{2}=1-a_{2},c_{2}=0\;;a_{3}-arbitary,b_{3}=0,c_{3}=1-a_{3}$

*



**Proof.** To check the associativity, it is enough to establish the associativity on the basis of elements $e_{1},e_{2}$ and $e_{3}$
$$e_{i}\circ \left(e_{j}\circ e_{k}\right)=\left(e_{i}\circ e_{j}\right)\circ e_{k},\;\mathrm{for}\;\mathrm{all}\;\;i,j,k=1,2,3\;$$By checking all the cases, we obtain the following equations
$$\begin{array}{c} b_{1}\left(1-b_{2}\right)=0\;\;\;b_{1}c_{2}=c_{1}\;\;\;b_{1}a_{2}+c_{1}=0\\ c_{1}\left(1-c_{3}\right)=0\;\;\;c_{1}b_{3}=b_{1}\;\;\;c_{1}a_{3}+b_{1}=0\\ a_{1}=1\;\;\;b_{1}=0\;\;\;c_{1}=0\\ a_{3}\left(1-a_{1}\right)=0\;\;\;a_{3}b_{1}=0\;\;\;a_{3}c_{1}=0\\ a_{2}\left(1-a_{1}\right)=0\;\;\;a_{2}b_{1}=0\;\;\;a_{2}c_{1}=0\\ b_{3}\left(1-a_{2}\right)=0\;\;\;b_{3}b_{2}=0\;\;\;b_{3}c_{2}=0\\ c_{2}\left(1-a_{3}\right)=0\;\;\;c_{2}b_{3}=0\;\;\;c_{3}c_{2}=0. \end{array}$$Solving these, we get
$$1.\;a_{1}=1,b_{1}=0,c_{1}=0\;;a_{2}=1,b_{2}=0,c_{2}=0\;;a_{3},b_{3},c_{3}-arbitary,b_{3}\neq 0.$$Hence, the corresponding operator $W_{1}$ has the following form:
$$W_{1}:\left\{\begin{array}{c} x_{1}^{{}^{\prime }}=x_{1}^{2}+x_{2}^{2}+a_{3}x_{3}^{2}+2x_{1}x_{2}+2x_{2}x_{3}+2x_{1}x_{3}\\ x_{2}^{{}^{\prime }}=b_{3}x_{3}^{2}\\ x_{3}^{{}^{\prime }}=c_{3}x_{3}^{2} \end{array}\right.$$
$$2.\;a_{1}=1,b_{1}=0,c_{1}=0\;;a_{2},b_{2},c_{2}-arbitary,c_{2}\neq 0\;;a_{3}=1,b_{3}=0,c_{3}=0.$$Hence, the corresponding operator $W_{1}$ has the following form:
$$W_{1}:\left\{\begin{array}{c} x_{1}^{{}^{\prime }}=x_{1}^{2}+a_{2}x_{2}^{2}+x_{3}^{2}+2x_{1}x_{2}+2x_{2}x_{3}+2x_{1}x_{3}\\ x_{2}^{{}^{\prime }}=b_{2}x_{2}^{2}\\ x_{3}^{{}^{\prime }}=c_{2}x_{2}^{2} \end{array}\right.$$

$$3.\;a_{1}=1,b_{1}=0,c_{1}=0\;;a_{2}-arbitary,b_{2}=1-a_{2},c_{2}=0\;;a_{3}-arbitary,b_{3}=0,c_{3}=1-a_{3}.$$

Hence, the corresponding operator $W_{1}$ has the following form:
$$W_{1}:\left\{\begin{array}{c} x_{1}^{{}^{\prime }}=x_{1}^{2}+a_{2}x_{2}^{2}+a_{3}x_{3}^{2}+2x_{1}x_{2}+2x_{2}x_{3}+2x_{1}x_{3}\\ x_{2}^{{}^{\prime }}=\left(1-a_{2}\right)x_{2}^{2}\\ x_{3}^{{}^{\prime }}=\left(1-a_{3}\right)x_{3}^{2} \end{array}\right.$$ □

### Dynamics of $W_{1}$

In this subsection, we are going to investigate the dynamics of a QSO corresponding to associative genetic algebra $A_{1}$. Let us study the dynamics of $W_{1}$ according to the different cases described in Theorem 2.

According to part (i), $W_{1}$ has the following form
$$W_{1}:\left\{\begin{array}{c} x_{1}^{{}^{\prime }}=x_{1}^{2}+x_{2}^{2}+a_{3}x_{3}^{2}+2x_{1}x_{2}+2x_{2}x_{3}+2x_{1}x_{3}\\ x_{2}^{{}^{\prime }}=b_{3}x_{3}^{2}\\ x_{3}^{{}^{\prime }}=c_{3}x_{3}^{2} \end{array}\right.$$

If $x_{3}=0$, then $W_{1}\left(x_{1},x_{2},0\right)=\left(1,0,0\right)$. If $x_{3}\neq 0\Rightarrow 0<x_{3}\leq 1$, then $x_{3}^{\left(1\right)}=C_{3}x_{3}^{2},$$\Rightarrow \;x_{3}^{\left(n\right)}=C_{3}^{2^{n}-1}x_{3}^{2^{n}}$. Now, if $c_{3}<1\Rightarrow \;\;x_{3}^{\left(n\right)}\to 0$ and $x_{2}^{\left(1\right)}=b_{3}x_{3}^{\left(2\right)}\Rightarrow x_{2}^{\left(n\right)}=b_{3}c_{3}^{2^{n-1}-1}x_{3}^{2^{n-1}}\Rightarrow x_{2}^{\left(n\right)}\to 0$. So, $x_{1}^{\left(n\right)}+x_{2}^{\left(n\right)}+x_{3}^{\left(n\right)}=1\Rightarrow x_{1}^{\left(n\right)}=1$. Thus, $W_{1}^{\left(n\right)}\left(x\right)\to \left(1,0,0\right)$. If $c_{3}=1,\Rightarrow a_{3}=b_{3}=0$, which gives $W_{1}\left(0,0,1\right)=\left(0,0,1\right)$. Therefore, $W_{1}\left(x_{1},x_{2},x_{3}\right)=\left(1-x_{3}^{2},0,x_{3}^{2}\right)$. If $x_{3}<1\Rightarrow \;\;W_{1}^{\left(n\right)}\left(x\right)\to \left(1,0,0\right)$.

According to part (ii), $W_{1}$ is represented as follows
$$W_{1}:\left\{\begin{array}{c} x_{1}^{{}^{\prime }}=x_{1}^{2}+a_{2}x_{2}^{2}+x_{3}^{2}+2x_{1}x_{2}+2x_{2}x_{3}+2x_{1}x_{3}\\ x_{2}^{{}^{\prime }}=b_{2}x_{2}^{2}\\ x_{3}^{{}^{\prime }}=c_{2}x_{2}^{2} \end{array}\right.$$

If $x_{2}=0$, then $W_{1}\left(x_{1},0,x_{3}\right)=\left(1,0,0\right)$. If $x_{2}\neq 0\Rightarrow 0<x_{2}\leq 1$, then $x_{2}^{\left(1\right)}=b_{2}x_{2}^{2},\Rightarrow x_{2}^{\left(n\right)}=b_{2}^{2^{n}-1}x_{2}^{2^{n}}$. Now, if $b_{2}<1\Rightarrow \;\;x_{2}^{\left(n\right)}\to 0$ and, $x_{3}^{\left(1\right)}=c_{2}x_{2}^{\left(2\right)}\Rightarrow x_{3}^{\left(n\right)}=c_{2}b_{2}^{2^{n-1}-1}x_{3}^{2^{n-1}}\Rightarrow x_{3}^{\left(n\right)}\to 0$. So, $x_{1}^{\left(n\right)}+x_{2}^{\left(n\right)}+x_{3}^{\left(n\right)}=1\Rightarrow x_{1}^{\left(n\right)}=1$. Thus, $W_{1}^{\left(n\right)}\left(x\right)\to \left(1,0,0\right)$. If $b_{2}=1,\Rightarrow a_{2}=b_{2}=0$, which gives $W_{1}\left(0,1,0\right)=\left(0,1,0\right)$. Therefore, $W_{1}\left(x_{1},x_{2},x_{3}\right)=\left(1-x_{2}^{2},x_{2}^{2},0\right)$. If $x_{2}<1\Rightarrow \;\;W_{1}^{n}\left(x\right)\to \left(1,0,0\right)$.

By part (iii), $W_{1}$ is given by
$$W_{1}:\left\{\begin{array}{c} x_{1}^{{}^{\prime }}=x_{1}^{2}+a_{2}x_{2}^{2}+a_{3}x_{3}^{2}+2x_{1}x_{2}+2x_{2}x_{3}+2x_{1}x_{3}\\ x_{2}^{{}^{\prime }}=\left(1-a_{2}\right)x_{2}^{2}\\ x_{3}^{{}^{\prime }}=\left(1-a_{3}\right)x_{3}^{2} \end{array}\right.$$

If $a_{2}=1,a_{3}=1$, then $W_{1}\left(x_{1},x_{2},x_{3}\right)\to \left(1,0,0\right)$. If $a_{3}=1,0\neq a_{2}<1$, then $x_{2}^{1}=\left(1-a_{2}\right)x_{2}^{2}\Rightarrow x_{2}^{\left(n\right)}={\left(1-a_{2}\right)}^{2^{n}-1}x_{2}^{2^{n}}$ because $a_{2}<1\Rightarrow x_{2}^{\left(n\right)}\to 0$. Additionally, $x_{3}^{\left(n\right)}=0\Rightarrow x_{1}^{\left(n\right)}\to$ 1. Therefore, $W_{1}^{\left(n\right)}\left(x\right)\to \left(1,0,0\right)$. If $a_{2}=1,0\neq a_{3}<1$, then $x_{3}^{1}=\left(1-a_{3}\right)x_{2}^{2}\Rightarrow x_{3}^{n}={\left(1-a_{3}\right)}^{2^{n}-1}x_{3}^{2^{n}}$ because $a_{3}<1\Rightarrow x_{3}^{\left(n\right)}\to 0$. Additionally, $x_{2}^{\left(n\right)}=0\Rightarrow x_{1}^{\left(n\right)}\to$ 1. Therefore, $W_{1}^{\left(n\right)}\left(x\right)\to \left(1,0,0\right)$. If $a_{2}<1,a_{3}<1$$x_{2}^{1}=\left(1-a_{2}\right)x_{2}^{2}$, which gives $x_{2}^{\left(n\right)}={\left(1-a_{2}\right)}^{2^{n}-1}x_{2}^{2^{n}}\Rightarrow x_{2}^{\left(n\right)}\to 0$, $x_{3}^{1}=\left(1-a_{3}\right)x_{2}^{2}$, which gives $x_{3}^{\left(n\right)}={\left(1-a_{3}\right)}^{2^{n}-1}x_{3}^{2^{n}}\Rightarrow x_{3}^{\left(n\right)}\to 0$$\Rightarrow x_{1}^{\left(n\right)}=1.$ Therefore, $W_{1}^{\left(n\right)}\left(x\right)\to \left(1,0,0\right)$. If $a_{2}=0\Rightarrow x_{2}^{{}^{\prime }}=x_{2}^{2}\Rightarrow x_{2}=0,1$ or $a_{3}=0\Rightarrow x_{3}^{{}^{\prime }}=x_{3}^{2}\Rightarrow x_{3}=0,1$.

Hence, we can formulate the following theorem.

**Theorem 3.** 
*Let $W_{1}$ be a QSO whose genetic algebra $A_{1}$ is associative, then $W_{1}$ is regular, moreover one has*

$$W_{1}^{\left(n\right)}\left(x\right)\to e_{1},\;\;\mathrm{for}\;\mathrm{every}\;\;x\in S^{2}.$$



## 5. Character

In this section, we characterize all characters of genetic algebras. Let *A* be a genetic algebra. Let us recall that a *character* of *A* is a linear functional on *A* with
h(x∘y)=h(x)h(y)∀x,y∈A.

We notice that the functional
$$h\left(x\right)=x_{1}+x_{2}+x_{3}$$
is a trivial character for any genetic algebra. Therefore, we are interested to find other nontrivial characters $A_{1}$.

**Theorem 4.** 
*Let us consider algebra $A_{1}$. Then, the following statements hold.*
*(i)* 
*If $c_{1}=c_{2}=0,c_{3}\neq 0$, then $h\left(x\right)=c_{3}x_{3}$ is a character;*
*(ii)* 
*If $b_{1}=b_{3}=0,b_{2}\neq 0$, then $h\left(x\right)=b_{2}x_{2}$ is a character;*
*(iii)* 
*otherwise, there is only a trivial character.*



**Proof.** Let $h\left(x\right)=h_{1}x_{1}+h_{2}x_{2}+h_{3}x_{3}$ be a linear functional, where $x=(x_{1},x_{2},x_{3})$. To check h is a character, it is enough to verify
(18)$$h\left(e_{i}\circ e_{j}\right)=h\left(e_{i}\right)h\left(e_{j}\right)\;\;\mathrm{for}\;\mathrm{all}\;\;i,j=1,2,3.$$It is clear that $h\left(e_{i}\right)=h_{i}$; then, checking (Equation 18) yields
(19)$$\begin{array}{c} a_{1}h_{1}+b_{1}h_{2}+c_{1}h_{3}=h_{1}^{2} \end{array}$$
(20)$$\begin{array}{c} h_{1}=h_{1}h_{2}\Rightarrow h_{1}\left(1-h_{2}\right)=0 \end{array}$$
(21)$$\begin{array}{c} h_{1}=h_{1}h_{3}\Rightarrow h_{1}\left(1-h_{3}\right)=0 \end{array}$$
(22)$$\begin{array}{c} a_{2}h_{1}+b_{2}h_{2}+c_{2}h_{3}=h_{2}^{2} \end{array}$$
(23)$$\begin{array}{c} h_{1}=h_{2}h_{3} \end{array}$$
(24)$$\begin{array}{c} a_{3}h_{1}+b_{3}h_{2}+c_{3}h_{3}=h_{3}^{2} \end{array}$$Now we want to solve these equations. Consider several cases.Case I: $h_{1}=0$, then $h_{2}h_{3}=0$.Sub-case $I_{1}$: Assume that $h_{2}=0,h_{3}\neq 0$.Then, from the above given equations, we find$c_{1}h_{3}=0\Rightarrow c_{1}=0$ and $c_{2}h_{3}=0\Rightarrow c_{2}=0$. Moreover, $h_{3}^{2}-c_{3}h_{3}=0\Rightarrow h_{3}=c_{3}\neq 0$Hence, $h\left(x\right)=c_{3}x_{3},\;\;c_{3}\neq o,c_{1}=c_{2}=0$.Sub-case $I_{2}$: $h_{2}\neq 0,h_{3}=0$

$b_{1}h_{2}=0\Rightarrow b_{1}=0$



$b_{3}h_{2}=0\Rightarrow b_{3}=0$



$h_{2}^{2}-b_{2}h_{2}=0\Rightarrow h_{2}=b_{2}\neq 0$

Thus, $h\left(x\right)=b_{2}x_{2},\;\;b_{2}\neq o,b_{1}=b_{3}=0$.If $h_{1}\neq 0,h_{2}=1,h_{3}=1$, then we get the trivial derivation. □

**Remark 2.** 
*It is worth mentioning that the characters of Lotka–Volterra and other kinds of genetic algebras have been investigated in [40,44].*


## 6. Derivations

In this section, we are going to describe derivations of genetic algebras associated with $\xi^{\left(a\right)}$-QSOs. We recall that a *derivation* on algebra (A,∘) is a linear mapping D:A→A such that D(u∘v)=D(u)∘v+u∘D(v) for all u,v∈A. It is clear that D≡0 is also a derivation, and such a derivation is called a *trivial* one. It is important to know whether the given algebra possesses a nontrivial derivation. Notice that a genetic interpretation of derivations was discussed in [35].

Let *A* be a genetic algebra associated with $W_{1}$. Its table of multiplication is given in Case 1. It is well known that *d* is a derivation if and only if
(25)$$d\left(e_{i}\circ e_{j}\right)=d\left(e_{i}\right)\circ e_{j}+e_{i}\circ d\left(e_{j}\right)$$ To describe derivations of the algebra *A*, we check the validity of (Equation 25). Assume that
(26)$$D\left(e_{i}\right)=\sum_{j=1}^{3}d_{i,j}e_{j},\;\;\;i\in 1,2,3$$
for some matrix $\left(d_{ij}\right)$. Then, we obtain the following system of equations:
{(27a)a1d11+b1d21+c1d31=2(a1d11+d12+d13)(27b)a1d12+b1d22+c1d32=2b1d11(27c)a1d13+b1d23+c1d33=2c1d11(27d)a2d11+b2d21+c2d31=2(d21+a2d22+d23)(27e)a2d12+b2d22+c2d32=2b2d22(27f)a2d13+b2d23+c2d33=2c2d22(27g)a3d11+b3d21+c3d31=2(d31+a2d32+d33)(27h)a3d12+b3d22+c3d32=2b3d33(27i)a3d13+b3d23+c3d33=2c3d33(27j)a2d12+d13+a1d21+d22+d23=0(27k)d12=b2d12+b1d21(27l)d13=c2d12+c1d21(27m)a3d13+d12+a1d31+d32+d33=0(27n)d12=b1d31+b3d13(27o)d13=c2d13+c1d31(27p)d21+d22+a3d23+d31+a3d32+d33=d11(27q)d12=b3d23+b2d32(27r)d13=c3d23+c2d32
In what follows, for the sake of simplicity, we restrict ourselves to case $c_{i}=0\;\forall i\in \left\{1,2,3\right\}$. In this case, the system is reduced to
{(28a)b1d21=a1d11+2d12+2d13(28b)a1d12+b1d22=2b1d11(28c)b1d23=0(28d)a2d11+b2d21=2(d21+a2d22+d23)(28e)a2d12=2b2d22(28f)b2d23=0(28g)a3d11+b3d21=2(d31+a2d32+d33)(28h)a3d12+b3d22=2b3d33(28i)b3d23=0(28j)a2d12+d13+a1d21+d22+d23=0(28k)(b2−1)d12+b1d21=0(28l)a3d13+d12+a1d31+d32+d33=0(28m)d12=b1d31+b3d13(28n)d21+d22+a3d23+d31+a3d32+d33=d11(28o)d12=b3d23+b2d32(28p)d13=0

Let us consider several cases:

**Case 1:** Assume that $b_{i}=0,i\in \left\{1,2,3\right\}$ which means $a_{i}=1,i\in \left\{1,2,3\right\}$; then, from the above equations, we obtain
$$\begin{array}{c} d_{11}=0\;\;d_{12}=0\;\;d_{13}=0\\ d_{21}+d_{22}+d_{23}=0\\ d_{31}+d_{32}+d_{33}=0 \end{array}$$
which yields
$$D=\left[\begin{array}{ccc} 0 & 0 & 0\\ \alpha & \beta & -\alpha -\beta \\ \gamma & \delta & -\gamma -\delta \end{array}\right],\;\alpha ,\beta ,\gamma ,\delta \;\mathrm{is}\;\mathrm{arbitrary}.$$

**Case 2:** Assume that $b_{i}=0,\forall i\in \left\{1,2,3\right\}$; then, from the above equations, one finds
$$\begin{array}{c} d_{13}=0,\;\;d_{23}=0,\;\;d_{32}=\frac{d_{12}}{b_{2}},\\ d_{31}=\frac{d_{12}}{b_{1}},\;\;d_{22}=\frac{a_{2}}{b_{1}}d_{12},\;\;d_{21}=\frac{a_{2}}{b_{1}}d_{12},\\ d_{11}=\frac{d_{12}}{2}\left\{\frac{a_{1}}{b_{1}}+\frac{a_{2}}{b_{2}}\right\},\;\;d_{33}=\frac{d_{12}}{2}\left\{\frac{a_{3}}{b_{3}}+\frac{a_{2}}{b_{2}}\right\}. \end{array}$$
From $a_{3}d_{13}+d_{12}+a_{1}d_{31}+d_{32}+d_{33}=0$, substituting values of $d_{31},d_{32},d_{33}$, we get
$$d_{12}\left\{\frac{a_{3}}{b_{3}}+\frac{a_{2}}{b_{2}}\right\}=-2\left\{1+\frac{a_{1}}{b_{1}}+\frac{1}{b_{2}}\right\}d_{12}.$$
If $d_{12}\neq 0$, because R.H.S is a negative number, then L.H.S must be negative, which is impossible, so $d_{12}=0$. This implies that $d_{ij}=0,\forall i,j\in \left\{1,2,3\right\}$. In this case, we have only the trivial derivation. 

**Case 3:** Assume that $b_{1}\neq 0,b_{2}=0,b_{3}=0$ which means $a_{1}=1,a_{2}=1$; then, from the above equations, we get
$$\begin{array}{c} d_{13}=0,\;\;d_{23}=0,\;\;d_{12}=0,\\ d_{31}=0,\;\;d_{21}=0\;\;,d_{22}=0,\\ d_{33}=-d_{32},\;\;d_{11}=0 \end{array}$$
which yields
$$D=\left[\begin{array}{ccc} 0 & 0 & 0\\ 0 & 0 & 0\\ 0 & \beta & -\beta \end{array}\right],\;\beta \;\mathrm{is}\;\mathrm{arbitrary}.$$

**Case 4:** Assume that $b_{1}=0,b_{2}\neq 0,b_{3}=0$; this means $a_{1}=1,a_{3}=1$. Then, using the same argument, we obtain a nontrivial derivation given by
$$D=\left[\begin{array}{ccc} 0 & 0 & 0\\ 0 & 0 & 0\\ \beta & 0 & -\beta \end{array}\right],\;\beta \;\mathrm{is}\;\mathrm{arbitrary}.$$

**Case 5:** Assume that $b_{1}=0,b_{2}=0,b_{3}\neq 0$, which means $a_{1}=1,a_{2}=1$. In this case, we need to examine the system
$$\begin{array}{c} d_{13}=0,\;\;d_{23}=0,\;\;d_{12}=0,\;\;d_{11}=0,\\ d_{21}=-d_{22},\;\;d_{33}=\frac{d_{22}}{2},\\ d_{31}=-d_{32}-\frac{d_{22}}{2}, \end{array}$$
which implies
$$D=\left[\begin{array}{ccc} 0 & 0 & 0\\ -\alpha & \alpha & 0\\ -\frac{\alpha }{2}-\beta & \beta & \frac{\alpha }{2} \end{array}\right],\;\alpha ,\beta \;\mathrm{is}\;\mathrm{arbitrary}.$$

**Case 6:** Assume that $b_{1}=0,b_{2}\neq 0,b_{3}\neq 0$; here $a_{1}=1$, hence
$$\begin{array}{c} d_{13}=0,\;\;d_{23}=0,\;\;d_{12}=0,\\ d_{32}=0,\;\;d_{21}=0\;\;,d_{22}=0,\\ d_{33}=-d_{31},\;\;d_{11}=0 \end{array}$$
which gives
$$D=\left[\begin{array}{ccc} 0 & 0 & 0\\ 0 & 0 & 0\\ \beta & 0 & -\beta \end{array}\right],\;\beta \;\mathrm{is}\;\mathrm{arbitrary}.$$

**Case 7:** Assume that $b_{1}\neq 0,b_{2}=0,b_{3}\neq 0$; this means $a_{2}=1$. Then, using the same argument, we obtain a nontrivial derivation given by
$$D=\left[\begin{array}{ccc} 0 & 0 & 0\\ 0 & 0 & 0\\ 0 & \beta & -\beta \end{array}\right],\;\beta \;\mathrm{is}\;\mathrm{arbitrary}.$$

**Case 8:** Assume that $b_{1}\neq 0,b_{2}\neq 0,b_{3}=0$, then we obtain $d_{ij}=0,\forall i,j\in \left\{1,2,3\right\}$. Hence, in this case, there is only a trivial derivation.

Now let us finalize the obtained results.

**Theorem 5.** *Let $A_{1}$ be the genetic algebra generated by $W_{1}$* (Equation 15) *with $c_{i}=0,\forall i\in \left\{1,2,3\right\}$. Then, the following statements hold.*
*(i)* *If all $b_{i}\neq 0,i\in \left\{1,2,3\right\}$ or $b_{1}\neq 0,b_{2}\neq 0,b_{3}=0$, then there is only a trivial derivation.**(ii)* *If $b_{1}\neq 0,b_{2}=0,b_{3}=0$ or $b_{1}\neq 0,b_{2}=0,b_{3}\neq 0$, then there is a nontrivial derivation given by*$$D=\left[\begin{array}{ccc} 0 & 0 & 0\\ 0 & 0 & 0\\ 0 & \beta & -\beta \end{array}\right]\;\beta \;\mathrm{is}\;\mathrm{arbitrary}.$$*(iii)* *If $b_{1}=0,b_{2}\neq 0,b_{3}=0$ or $b_{1}=0,b_{2}\neq 0,b_{3}\neq 0$, then there is a nontrivial derivation given by*$$D=\left[\begin{array}{ccc} 0 & 0 & 0\\ 0 & 0 & 0\\ \beta & 0 & -\beta \end{array}\right]\;\beta \;\mathrm{is}\;\mathrm{arbitrary}.$$*(iv)* *If $b_{1}=0,b_{2}=0,b_{3}\neq 0$, then there is a nontrivial derivation given by*$$D=\left[\begin{array}{ccc} 0 & 0 & 0\\ -\alpha & \alpha & 0\\ -\frac{\alpha }{2}-\beta & \beta & \frac{\alpha }{2} \end{array}\right]\;\alpha ,\beta \;is\;arbitary.$$*(v)* *If $b_{i}=0,\forall i\in \left\{1,2,3\right\}$, then there is a nontrivial derivation given by*$$D=\left[\begin{array}{ccc} 0 & 0 & 0\\ \alpha & \beta & -\alpha -\beta \\ \gamma & \delta & -\gamma -\delta \end{array}\right]\;\alpha ,\beta ,\gamma ,\delta \;is\;arbitary.$$

## 7. Dynamics of Some $\xi^{\left(a\right)}$-QSOs

This section is devoted to the investigation of the dynamical behavior of $\xi^{\left(a\right)}$-QSOs. We concentrate on the investigation of $V_{1},V_{2},\ldots ,V_{9}$ operators given in Section 2. Using the argument of Theorem (1), we can establish that $V_{1}$ is conjugate to $V_{2},V_{6}$; $V_{4}$ is conjugate to $V_{8},V_{9}$, and $V_{3}$ is conjugate to $V_{5},V_{7}$. Therefore, we concentrate on the investigation of the $V_{1}$, $V_{4}$ and $V_{7}$ operators, which will be studied separately. Furthermore, in order to provide a visual representation of the behavior of the considered class of $\xi^{\left(a\right)}$-quadratic stochastic operators (QSOs), we present accompanying images that illustrate their dynamics. These images aim to aid in understanding and interpreting the behavior of the operators in a graphical manner.

### 7.1. Dynamics of $V_{1}$

Now, we are going to study the dynamics of $V_{1}$. The dynamics of $V_{1}$ depend on the value of the parameter α. For this reason, we are going to consider three cases; namely, when α=1,α=0, and 0<α<1.

Let $l_{z}=\left\{\left(x,y,0\right),x,y\geq 0,x+y=1\right\}$.

**Proposition 1.** 
*The following statement holds for $V_{1}$:*
*1.* 
*If $x^{\left(0\right)}\notin \mathrm{Fix}\left(V_{1}\right)$ be any initial point, then $V_{1}\left(x^{\left(0\right)}\right)\in l_{z=0}.$*
*2.* 
*The line $l_{z=0}$ is invariant.*



**Proof.** The proof is straightforward. □

Let us assume that α=1. Then, $V_{1}$ has the following form:
$$V_{1}:\left\{\begin{array}{c} x^{\prime }=x^{2}+2xy+2xz+2yz\\ y^{\prime }=y^{2}+z^{2}\\ z^{\prime }=0 \end{array}\right.$$ Due to Proposition (1), it is enough to study the dynamic of $V_{1}$ on the line $l_{z=0}.$ Hence, the second coordinate becomes $y^{\prime }=y^{2}.$ So, the fixed points of $V_{1}$ when α=1 are (1,0,0) and (0,1,0). Because $y^{\prime }=y^{2}$, then the sequence $\left\{y^{\left(n\right)}\right\}$ is decreasing and bounded; this implies that $y^{\left(n\right)}\to 0.$ Hence, $x^{\left(n\right)}\to 1-y^{\left(n\right)}.$ Thus, $\omega (x^{\left(0\right)})=\left(1,0,0\right),$ as shown in Figure 1.

Assume that α=0; then $V_{1}$ becomes
$$V_{1}:\left\{\begin{array}{c} x^{\prime }=\left(y^{2}+z^{2}\right)+2xy+2xz+2yz\\ y^{\prime }=x^{2}\\ z^{\prime }=0 \end{array}\right.$$ To find the fixed point of $V_{1}$ in this case, we use the second coordinate and x=1−y. Hence, $y={\left(1-y\right)}^{2}$, which is equivalent to $y^{2}-3y+1=0.$ The solution of the last equation is $y=\frac{3\pm \sqrt{5}}{2}.$ However, $\frac{3+\sqrt{5}}{2}>1.$ Hence, the fixed point in this case $\mathrm{Fix}\left(V_{1}\right)=\left\{\left(\frac{-1+\sqrt{5}}{2},\frac{3-\sqrt{5}}{2},0\right)\right\}.$ In this case we have also periodic points. From the second coordinate, one has $y^{\prime \prime }={\left(1-{\left(1-y\right)}^{2}\right)}^{2}$, which is equivalent to $y^{\prime \prime }={\left(2y-y^{2}\right)}^{2}.$ So, $y^{\prime \prime }=y$ if and only if $y\in \{0,\frac{3-\sqrt{5}}{2},1\}.$ So, the periodic points of $V_{1}$ when α=0 are as follows:
$$\left\{e_{1},e_{2},\left(\frac{-1+\sqrt{5}}{2},\frac{3-\sqrt{5}}{2},0\right)\right\}.$$ Define the function $f\left(y\right)={\left(1-y\right)}^{2}$, this function is decreasing on [0,1]. Consider $g\left(y\right):=f\left(f\left(y\right)\right)-y=y^{4}-4y^{3}+4y^{2}-y.$ After simple calculations, one has g(y)<0 when $0\leq y<\frac{3-\sqrt{5}}{2}$, and g(y)>0 when $\frac{3-\sqrt{5}}{2}<y\leq 1$.

Assume that $x^{\left(0\right)}\in l_{z=0}$ is any initial point. If $x^{\left(0\right)}$ is chosen such that $y^{\left(0\right)}<\frac{-1+\sqrt{5}}{2}$, because f(y) is decreasing, then $y^{\left(1\right)}>\frac{-1+\sqrt{5}}{2}$. Then, the trajectory implies that the sequence $\frac{-1+\sqrt{5}}{2}<\left\{y^{\left(2k\right)}\right\}\leq 1$ and the sequence $0\leq \left\{y^{\left(2k+1\right)}\right\}<\frac{-1+\sqrt{5}}{2}.$ One can find that the sequence $\left\{y^{\left(2k+1\right)}\right\}$ is decreasing. Hence, $y^{\left(2k+1\right)}\to 0$, consequently, $x^{\left(2k+1\right)}\to 1.$ Additionally, the sequence $\left\{y^{\left(2k\right)}\right\}$ is increasing. Hence, $y^{\left(2k\right)}\to 1$, consequently, $x^{\left(2k+1\right)}\to 0.$ Thus,
$$\omega \left(x^{\left(0\right)}\right)=\left\{e_{1},e_{2}\right\}.$$

From Figure 2, one can see that the trajectory jumps between the periodic points $\left\{e_{1},e_{2}\right\}$.

Now, consider 0<α<1. Then, to find the fixed points, we shall solve the following systems
$$\begin{array}{ccc} \alpha x^{2}+\left(1-\alpha \right){\left(1-x\right)}^{2}+2x\left(1-x\right) & = & x\\ y^{2}+\left(2\alpha -2\right)y+1-\alpha & = & y \end{array}$$ From $y^{2}+\left(2\alpha -2\right)y+1-\alpha =y$, one finds $y=\frac{3-2\alpha -\sqrt{4\alpha^{2}-8\alpha +5}}{2}.$ Hence, $x=\frac{\left(2\alpha -1\right)+\sqrt{4\alpha^{2}-8\alpha +5}}{2}.$ So, $\left(\frac{\left(2\alpha -1\right)+\sqrt{4\alpha^{2}-8\alpha +5}}{2},\frac{3-2\alpha -\sqrt{4\alpha^{2}-8\alpha +5}}{2},0\right)$ is a fixed point. Define the function
$$h\left(y\right)=y^{2}+\left(2\alpha -2\right)y+1-\alpha .$$ This function increases for any y≥1−α and decreases for any y≤1−α.

Let us denote $\Delta =4\alpha^{2}-8\alpha +5$. The following result is well known [48] (see also [49]).

**Theorem 6.** 
*The following statements hold:*
*(i)* 
*If 0<Δ<4, then all the trajectories of V converge to the fixed point.*
*(ii)* 
*If 4<Δ<5, then there exist two periodic points of V, and all trajectories go to them except for the fixed point.*



Now we are going to clarify under which conditions of α we can explicitly find the fixed and periodic points respectively.

(i)Assume that 0<Δ<4, then
$$4\alpha^{2}-8\alpha +5<4\Rightarrow 4\alpha^{2}-8\alpha +1<0\Rightarrow 1-\frac{\sqrt{3}}{2}<\alpha \leq 1.$$The unique fixed point is given by
$$f_{1}=\left(\frac{\left(2\alpha -1\right)+\sqrt{4\alpha^{2}-8\alpha +5}}{2},\frac{3-2\alpha -\sqrt{4\alpha^{2}-8\alpha +5}}{2},0\right).$$(ii)Let us assume that 4<Δ<5, then, keeping in view the above calculations, we have
$$0<\alpha \leq 1-\frac{\sqrt{3}}{2}.$$In this case, *V* has two periodic points. To find them, we need to solve
$$y^{4}+\left(4\alpha -4\right)y^{3}+\left(4a^{2}-8a+4\right)y^{2}-y+a-a^{2}=0.$$The solutions of this equation are
$$y=\frac{3-2\alpha \pm \sqrt{4\alpha^{2}-8\alpha +5}}{2},\;y=\frac{1-2\alpha \pm \sqrt{4\alpha^{2}-8\alpha +5}}{2}.$$Hence, two periodic points are given by
$$f_{2}=\left(\frac{1+2\alpha -\sqrt{4\alpha^{2}-8\alpha +1}}{2},\frac{\left(1-2\alpha \right)+\sqrt{4\alpha^{2}-8\alpha +1}}{2},0\right)$$
$$f_{3}=\left(\frac{1+2\alpha +\sqrt{4\alpha^{2}-8\alpha +1}}{2},\frac{\left(1-2\alpha \right)-\sqrt{4\alpha^{2}-8\alpha +1}}{2},0\right).$$We note that
$$f_{1}=\left(\frac{\left(2\alpha -1\right)+\sqrt{4\alpha^{2}-8\alpha +5}}{2},\frac{3-2\alpha -\sqrt{4\alpha^{2}-8\alpha +5}}{2},0\right)$$
is a fixed point of $V_{1}$.

Furthermore, the point
$$f_{4}=\left(\frac{\left(2\alpha -1\right)-\sqrt{4\alpha^{2}-8\alpha +5}}{2},\frac{3-2\alpha +\sqrt{4\alpha^{2}-8\alpha +5}}{2},0\right)$$
does not belong to the simplex $S^{2}$. Now, keeping in mind theorem 6, we can summarize the following result:

(i)If $0<\alpha \leq 1-\frac{\sqrt{3}}{2},$ then for any $x^{\left(0\right)}\in S^{2}$ we have $\omega \left(x^{\left(0\right)}\right)=\left\{f_{2},f_{3}\right\}$.(ii)If $1-\frac{\sqrt{3}}{2}<\alpha <1,$ then for any $x^{\left(0\right)}\in S^{2}$ one has $\omega \left(x^{\left(0\right)}\right)=\left\{f_{1}\right\}$.

**Remark 3.** 
*We stress that the dynamics of $V_{4}$ can be investigated by the same argument as $V_{1}$. Therefore, we leave this without going into detail.*


### 7.2. Dynamics of $V_{7}$

In this section, we are going to study the general properties of the operator $V_{7}.$ The finding of a fixed point depending on the parameter α is a difficult task. Hence, we are going to estimate the region of the fixed point.

**Proposition 2.** 
*The following statements hold for $V_{7}:$*
*(i)* $x^{\prime }+y^{\prime }\geq 1/3$.*(ii)* 
*If $z^{\left(0\right)}<\frac{1}{2},$ then the sequence $\left\{z^{\left(n\right)}\right\}$ is strictly increasing.*
*(iii)* 
*If $\alpha <\frac{1}{2}$ then $x^{\left(n\right)}>y^{\left(n\right)},$ if $\alpha >\frac{1}{2}$ then $x^{\left(n\right)}<y^{\left(n\right)},$ and if $\alpha =\frac{1}{2}$ then $x^{\left(n\right)}=y^{\left(n\right)}.$*



**Proof.** Consider $x^{\prime }+y^{\prime }=x^{2}+y^{2}+z^{2}$. Using the Lagrange Multilayer method, one has that the minimum value of the function $x^{2}+y^{2}+z^{2}$, subject to x+y+z=1 and x,y,z≥0, is $\frac{1}{3}.$ This implies that $x^{\prime }+y^{\prime }\geq \frac{1}{3}$.To prove (ii), let us take
$$z^{\prime }-z=2z\left(x+y\right)+2xy-z=z-2z^{2}+2xy.$$ It is not hard to show that $z\geq 2z^{2}$ in $\left[0,\frac{1}{2}\right]$, then $z-2z^{2}+2xy\geq 0.$ This implies that $z^{\prime }>z.$ Hence, the sequence $\left\{z^{\left(n\right)}\right\}$ is strictly increasing.For (iii), consider
$$x^{\left(n\right)}-y^{\left(n\right)}=\left(2\alpha -1\right){\left(x^{\left(n-1\right)}\right)}^{2}+\left(1-2\alpha \right){\left(y^{\left(n-1\right)}\right)}^{2}+\left(1-2\alpha \right){\left(z^{\left(n-1\right)}\right)}^{2}.$$ By (ii) we have that $z^{\left(n-1\right)}$ is going to be the maximum value of $\{x^{\left(n-1\right)},x^{\left(n-1\right)},z^{\left(n-1\right)}\}.$ This implies that, if $\alpha <\frac{1}{2}$, then
$$x^{\left(n\right)}-y^{\left(n\right)}=\left(2\alpha -1\right){\left(x^{\left(n-1\right)}\right)}^{2}+\left(1-2\alpha \right){\left(y^{\left(n-1\right)}\right)}^{2}+\left(1-2\alpha \right){\left(z^{\left(n-1\right)}\right)}^{2}>0$$
and if $\alpha >\frac{1}{2}$, then $x^{\left(n\right)}-y^{\left(n\right)}<0$, and if $\alpha =\frac{1}{2}$, then $x^{\left(n\right)}-y^{\left(n\right)}=0.$ □

### 7.3. The Dynamic of $V_{7}$ When α=1

In this subsection, we are going to study the dynamic of $V_{7}$ when α=1. Substituting α=1 in $V_{7}$, one has the following operator:
$$V_{7}:\left\{\begin{array}{c} x^{\prime }=x^{2}\\ y^{\prime }=\left(y^{2}+z^{2}\right)\\ z^{\prime }=2xy+2xz+2yz \end{array}\right.$$

**Theorem 7.** 
*The following hold true for $V_{7}$ when α=1:*
*(i)* 

$\mathrm{Fix}\left(V_{7}\right)=\left\{e_{1},e_{2},\left(0,\frac{1}{2},\frac{1}{2}\right)\right\}$

*(ii)* 
*If $x^{\left(0\right)}\notin \mathrm{Fix}\left(V_{7}\right)$ is any initial point, then $\omega \left(x^{\left(0\right)}\right)=\left\{\left(0,\frac{1}{2},\frac{1}{2}\right)\right\}.$*



**Proof.** To find the fixed point, we must solve the following system
$$\begin{array}{ccc} x^{2} & = & x\\ y^{2}+z^{2} & = & y\\ 2xy+2xz+2yz & = & z. \end{array}$$ Then, x∈{0,1}. If x=1, we have the fixed point $e_{1}.$ If x=0, then we use the fact x+y+z=1, which implies that z=1−y. Putting this value into the second equation of the above system yields $2y^{2}-3y+1=0.$ Hence, $y\in \{\frac{1}{2},1\}.$ If y=1, then we get the fixed point $e_{2}.$ If $y=\frac{1}{2}$, then $z=1-y=\frac{1}{2}.$ Consequently, we have the fixed point $\left(0,\frac{1}{2},\frac{1}{2}\right)$.To prove (ii), we note that the sequence $\left\{x^{\left(n\right)}\right\}$ is strictly decreasing and bounded. Hence, it converges to a fixed point, which is 0. Thus, it is enough to study the dynamic on the line $l_{x=0}.$ Define the function $k\left(y\right)=2y^{2}-2y+1.$ One can show that the last function is decreasing when $0\leq y\leq <\frac{1}{2}$ and increasing when $\frac{1}{2}\leq y\leq 1.$ Using (i) of Proposition (2), we get $y^{\prime }\geq \frac{1}{3}.$ Because k(y) is decreasing when $0\leq y\leq <\frac{1}{2}$, then $k\left(\left[\frac{1}{3},\frac{1}{2}\right]\right)\subset \left[\frac{1}{2},1\right].$ Because k(y) is increasing when $\frac{1}{2}\leq y\leq 1$, then $k\left(\left[\frac{1}{2},1\right]\right)\subset \left[\frac{1}{2},1\right].$ So, the dynamic of $V_{3}$ is reduced to the region when $y\in \left[\frac{1}{2},1\right]$ Now, consider $k\left(y\right)-y=2y^{2}-3y+1.$ It is easy to show that k(y)≤y in $\left[\frac{1}{2},1\right].$ Hence, the sequence $\left\{y^{\left(n\right)}\right\}$ is decreasing and bounded. Therefore, $y^{\left(n\right)}\to \frac{1}{2}.$ This implies that
$$\omega \left(x^{\left(0\right)}\right)=\left\{\left(0,\frac{1}{2},\frac{1}{2}\right)\right\}.$$ □

The following Figure 3 shows the dynamic of $V_{7}$ when α=1.

### 7.4. The Dynamic of $V_{7}$ When $\alpha =\frac{1}{2}$

In this section, we are going to study the dynamic of $V_{7}$ when $\alpha =\frac{1}{2}$. Substituting $\alpha =\frac{1}{2}$ in $V_{3}$, one has the following operator:
$$V_{7}:\left\{\begin{array}{c} x^{\prime }=x^{2}+y^{2}+z^{2}\\ y^{\prime }=x^{2}+y^{2}+z^{2}\\ z^{\prime }=2xy+2xz+2yz \end{array}\right.$$

**Theorem 8.** 
*The following hold true for $V_{7}$ when $\alpha =\frac{1}{2}:$*
*(i)* $\mathrm{Fix}\left(V_{7}\right)=\left\{\left(\frac{1}{2}-\frac{\sqrt{3}}{6},\frac{1}{2}-\frac{\sqrt{3}}{6},\frac{\sqrt{3}}{3}\right)\right\}$.*(ii)* 
*The line $l_{x=y}$ is invariant.*
*(iii)* 
*If $x^{\left(0\right)}\notin \mathrm{Fix}\left(V_{7}\right)$ is any initial point, then $\omega \left(x^{\left(0\right)}\right)=\left\{\left(\frac{1}{2}-\frac{\sqrt{3}}{6},\frac{1}{2}-\frac{\sqrt{3}}{6},\frac{\sqrt{3}}{3}\right)\right\}.$*



**Proof.** To find the fixed point, we shall solve the following system:
$$\begin{array}{ccc} \frac{1}{2}x^{2}+\frac{1}{2}y^{2}+\frac{1}{2}z^{2} & = & x\\ \frac{1}{2}x^{2}+\frac{1}{2}y^{2}+\frac{1}{2}z^{2} & = & y\\ 2xy+2xz+2yz & = & z \end{array}$$ Clearly, x=y and we use z=1−x−y; then, the first equation becomes $3x^{2}-2x+\frac{1}{2}=0$. The solutions of the last equation are $x=\frac{1}{2}-\frac{\sqrt{3}}{6},\;x=\frac{1}{2}+\frac{\sqrt{3}}{6}.$ The solution $x=\frac{1}{2}+\frac{\sqrt{3}}{6}$ is rejected because x+y≤1. Hence, we have the fixed point $\left(\frac{1}{2}-\frac{\sqrt{3}}{6},\frac{1}{2}-\frac{\sqrt{3}}{6},\frac{\sqrt{3}}{3}\right)$.The proof of (ii) is straightforward.To prove (iii), it is enough to study the trajectory on the line $l_{x=y}.$ To complete this task, define the function $m\left(x\right)=3x^{2}-2x+\frac{1}{2}.$ This function is decreasing when $0\leq x\leq \frac{1}{3}$ and increasing when $\frac{1}{3}\leq x\leq 1$ due to the fact x=y; this implies that $x\leq \frac{1}{2}.$ Additionally, from the fact $x+y>\frac{1}{3}$, this implies that $x\geq \frac{1}{6}.$ So, it is enough to study the dynamic when $\frac{1}{6}\leq x\leq \frac{1}{2}.$ One can show that $m\left(\left[\frac{1}{6},x^{*}\right]\right)\subset \left[x^{*},\frac{1}{4}\right]$ where $x^{*}=\frac{1}{2}-\frac{\sqrt{3}}{6}.$ Additionally, $m\left(\left[\frac{1}{3},\frac{1}{2}\right]\right)\subset \left[\frac{1}{6},\frac{1}{4}\right].$ Consider the function $m\left(m\left(x\right)\right)-x=27\;x^{4}-36\;x^{3}+15\;x^{2}-3\;x+\frac{1}{4}.$ One can see that this function is increasing when $x\in \left[\frac{1}{6},x^{*}\right]$ and decreasing when $x\in \left[x^{*},\frac{1}{4}\right].$ So, $x^{\left(2n\right)}\to x^{*}$ and $x^{\left(2n+1\right)}\to x^{*}.$ Hence, if $x^{\left(0\right)}\notin \mathrm{Fix}\left(V_{7}\right)$ is any initial point, then $\omega \left(x^{\left(0\right)}\right)=\left\{\left(\frac{1}{2}-\frac{\sqrt{3}}{6},\frac{1}{2}-\frac{\sqrt{3}}{6},\frac{\sqrt{3}}{3}\right)\right\}.$ □

The following Figure 4 is the dynamic of $V_{7}$ when $\alpha =\frac{1}{2}$.

**Remark 4.** 
*From the results, we infer that the considered operators are regular to the unique fixed point. This indicates whether these operators are contractions or not. It turns out that these operators are not contractions. Indeed, to verify this, one needs to check the condition [11]*

$$\max_{i_{1}.i_{2},k}\sum_{j=1}^{d}\left|p_{i_{1}k,j}-p_{i_{2}k,j}\right|<1$$

*One can check, for example, for $V_{7}$, that*

$$\sum_{j=1}^{3}\left|p_{1,1,1}-p_{2,1,1}\right|<1\Rightarrow 2<1\;\;\mathrm{is}a\mathrm{contradiction}$$

*which implies that $V_{7}$ is not a contraction. Up to now, there is no clear rigorous proof of the regularity of these operators in a general setting.*


## 8. Conclusions

In the current paper, we investigated the algebraic properties of the genetic algebras associated with $\xi^{\left(a\right)}$-QSOs. The associativity of these operators corresponding to partition $\xi_{5}$, along with their dynamics, were studied. The characters of these QSOs were described. We also fully characterized all derivations of such kinds of algebras. Finally, the regularity of the dynamics of $\xi^{\left(a\right)}$-QSOs were investigated. However, the study of the behavior of these operators in higher dimensional simplex still remains as an open problem. Further work could include generalization to other classes of QSOs; while the present paper focuses on a specific class of QSOs corresponding to the partition $\xi_{5}$, there are other partitions and classes that could be explored. Investigating the algebraic properties, dynamics, and behavior of these different classes could provide a more comprehensive understanding of QSOs as a whole.

## Figures and Tables

**Figure 1 entropy-25-00934-f001:**
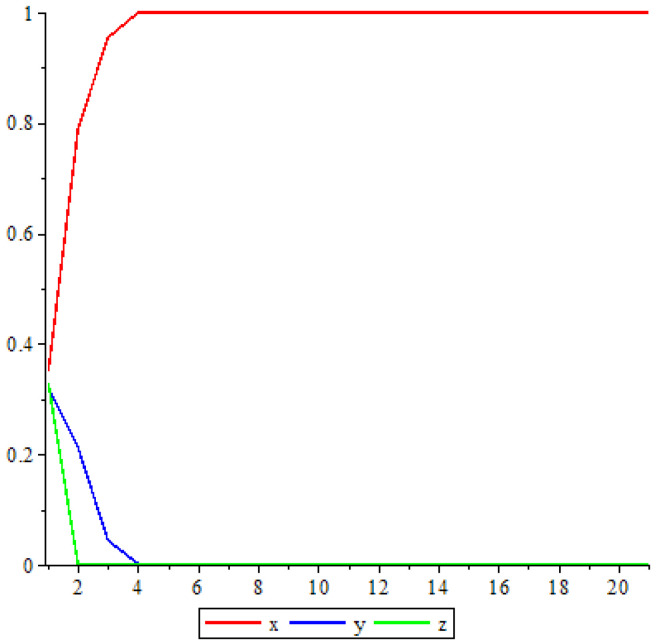
Trajectory when α=1.

**Figure 2 entropy-25-00934-f002:**
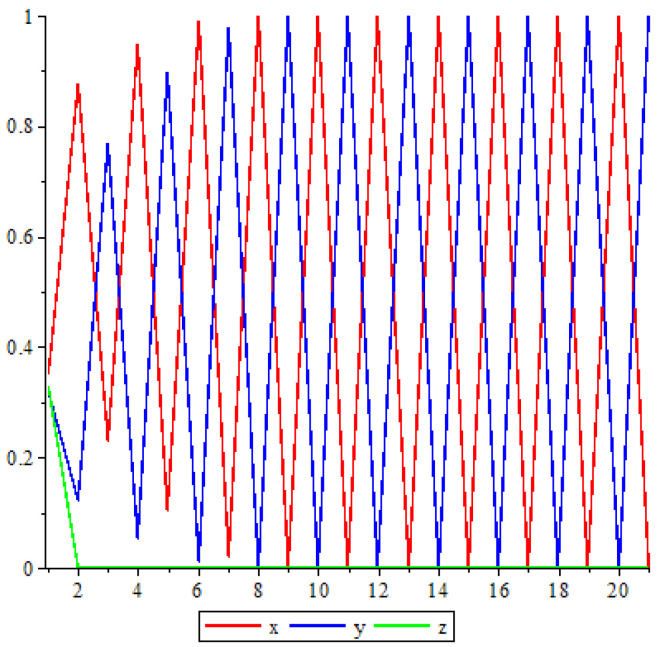
Trajectory when α=0.

**Figure 3 entropy-25-00934-f003:**
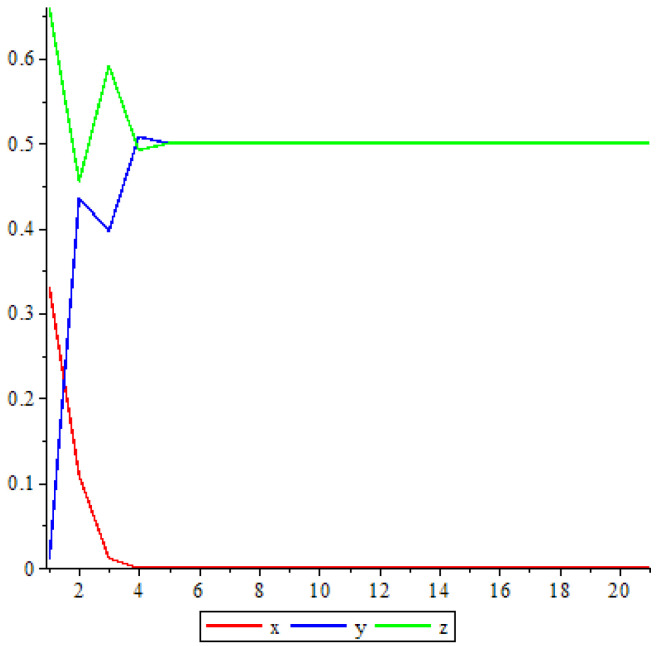
Trajectory when α=1.

**Figure 4 entropy-25-00934-f004:**
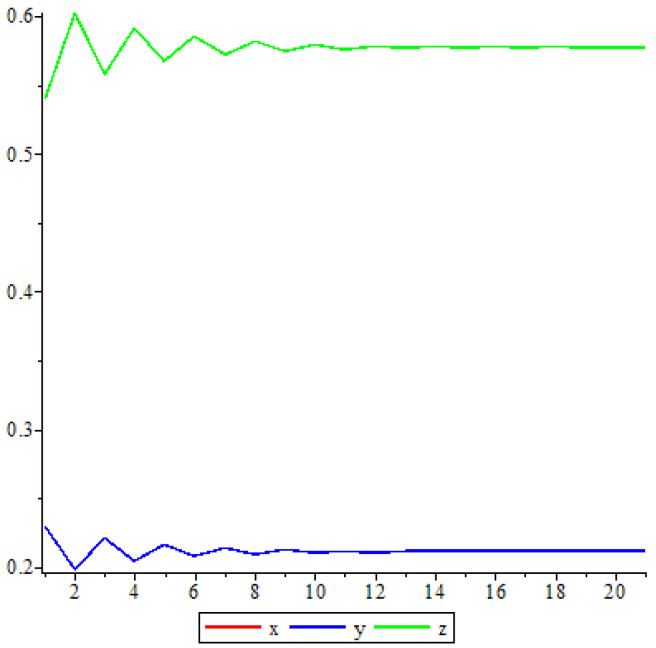
Trajectory when $\alpha =\frac{1}{2}$.

## Data Availability

Not applicable.

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
