# Peer review of "Genetic Algebras Associated with ξ(a)-Quadratic Stochastic Operators"

_entropy, 2023, doi:10.3390/e25060934_

Round 1
Reviewer 1 Report
In the peer-reviewed paper, the authors explore an algebraic approach to the analysis of essential properties of population genetics. Such algebras are referred to as "genetic algebras" and in many works the algebras are not associative. An important tool for analyzing the problems of population genetics are the quadratic stochastic operators (QSO) initiated by S. Bernstein in 1942 and developed in many subsequent works. Quadratic operators may define some algebraic structure (a genetic algebra) on some vector space with a simplex.
In this paper, the authors consider algebraic features of genetic algebras associated with a special class of QSO (\xi_(a)-QSO) on simplices that have not been systematically studied before. Associativity s of \xi_(a)-QSOs with their dynamics and a number of other algebraic properties are investigated.
The paper contributes to the study of population genetics and, in particular, provides new information about \xi_(a) - quadratic stochastic operators and their properties.
The text of the article is well structured, written clearly, the introduction provides the necessary information about the field of study and the methods used, the research topic is sufficiently motivated.
In my opinion, the paper can be accepted for publication in Entropy in present form..
Author Response
Dear Reviewer,
Thank you for your comments.
Best regards,
Farrukh Mukhamedov
Reviewer 2 Report
1. What is QSO in the abstract?
2. Abstract of the paper is very short. It should include the main outcomes of the paper.
3. All references cited in the introduction section must be in the same order. Such as [2] must be after [1] and so on.
4. The authors are advised to improve their introduction by adding the following recent work in the introduction section:
a) Chen Bifurcation, chaos, and circuit realisation of a new four-dimensional memristor system
b) Strong resonance bifurcations for a discrete-time prey–predator model
c) Dynamical Behaviors of a SIR Epidemic Model with Discrete Time
d) PUBLIC Information, Actual Intervention and Inflation Expectations
e) A Comparison Research on Dynamic Characteristics of Chinese and American Energy Prices
5. Definitions 2.1 and 2.2 are newly introduced or taken from the literature, if taken from the literature then must be a reference there.
6. There is no motivation in the paper.
7. In section 3 for the P3 partition, the authors represent 1,2,3 then why there is no (3,3) and (3,1)… and so on.
8. What is the main objective of this work?
9. From the figures what the authors have observed.
10. Conclusion should be revised and highlight the future direction for this work.
The authors are advised to check the whole manuscript for English editing, typos and punctuation.
Author Response
Dear Reviewer,
Thank you for your comments. We have fixed all raised issues.
Best regards,
Farrukh Mukhamedov

Reviewer 3 Report
Please, find the report attached.

Author Response

(The authors gave the same response as above.)

Round 2
Reviewer 2 Report
The authors have improved the manuscript in the revision. I recommend the paper for publication
The authors have improved the English of the paper.